# Germplasm Diversification in Citrus Orchards in a Mesothermal Climate in Brazil

Gabriel Maluf Napoleão [1,*], Sarita Leonel [1], Jackson Mirellys Azevedo Souza [2], Magali Leonel [3], Rafaelly Calsavara Martins [1], Caroline Pardine Cardoso [4], Marco Antonio Tecchio [1] and Jaime Duarte Filho [5]

[1] Department of Horticulture, School of Agriculture (FCA), São Paulo State University (UNESP), Botucatu 18610-034, São Paulo, Brazil; sarita.leonel@unesp.br (S.L.); rcalsavara@yahoo.com.br (R.C.M.); marco.a.tecchio@unesp.br (M.A.T.)

[2] Department of Agronomy, Federal University of Viçosa (UFV), Viçosa 36570-900, Minas Gerais, Brazil; jackson.m.souza@ufv.br

[3] Center for Tropical Root and Starches (CERAT), São Paulo State University (UNESP), Botucatu 18610-034, São Paulo, Brazil; magali.leonel@unesp.br

[4] Department of Biodiversity and Biostatistics, Institute of Biosciences, São Paulo State University (UNESP), Botucatu 18618-689, São Paulo, Brazil; caroline.pardine@unesp.br

[5] Regional Development Coordination, Department of Agriculture and Supply of the State of São Paulo, Botucatu 18607-050, São Paulo, Brazil; jaime.duarte@sp.gov.br

* Correspondence: gabrielmaluf275@gmail.com

**Abstract:** The limited scion-rootstock combinations used in sweet orange orchards in Brazil reduce the commercial potential of citrus and lead to greater susceptibility to pests and diseases. Aiming to provide germplasm diversification, the research evaluated the Folha Murcha (FM) and Charmute de Brotas (CB) orange trees grafted onto 'Rangpur' lime (RL) and 'Swingle' citrumelo (SC) rootstocks. The study was conducted in the southern center of the state of São Paulo, in the subtropical region of southeastern Brazil. The grafted trees were planted in September 2016. The field experiment was conducted over two seasons. The combinations were evaluated based on the duration in days and degree-days of the phenological intervals, growth and yield performance, alternate bearing, fruit ripening, and quality. The trees of the two scion cultivars grafted onto RL had the shortest crop cycle, the earliest fruit ripening, and the largest canopy volume. SC produced fewer alternate bearings with greater production efficiency. The CB/SC combination produced fruits with a better color index, higher total soluble solids content, and a higher technological index. This assessment can be useful in planning orchards for dual-purpose markets, such as table fruit and the juice industry.

**Keywords:** alternate bearing; canopy volume; *Citrus limonia* Osbeck; *Citrus sinensis* [L.] Osbeck; [*P. trifoliata* (L.) Raf × *Citrus paradisi* Macf.]; degree-days; fruit ripening; rootstock; scion

## 1. Introduction

Citrus orchards in São Paulo State are responsible for the largest volume of fruit produced in Brazil and are mainly focused on producing sweet oranges for freezing and concentrated juice for export [1]. Despite its leadership in this activity, Brazil is still not well known for its table citrus production. However, there has been growing interest in the production sector in the diversification of scion cultivars intended for whole-fruit consumption [2].

With the objective of shortening the off-season, the management of orchards has been adjusted to achieve a greater balance in the number of trees with early, intermediate, and late maturation. Planting different cultivars is another way to reduce phytosanitary risks and the impacts of climate change [3]. Using new cultivars can mitigate environmental risks and phytosanitary damage and add desirable characteristics, such as harvest staggering, higher productivity, and fruit quality, to meet market demands [4].

The most planted and commercialized sweet oranges in Brazil are the Pera cultivars, which are considered mid-season, and the Valencia, Natal, and Folha Murcha cultivars, which are considered late-ripening, among which the late-ripening cultivars are the most cultivated for industrial processing [5]. In the 2021/2022 orange harvest in the citrus belt of São Paulo/southwest Minas Gerais, the Pera cultivar represented 35.6% of the productive trees in the region, whereas the Valencia and Folha Murcha cultivars represented 33.5%, followed by the Natal cultivar at 11.25% [6].

The Folha Murcha cultivar (*Citrus sinensis* [L.] Osbeck) was selected from Araruama, Rio de Janeiro, Brazil. Its origin was probably due to the spontaneous mutation of 'Valencia,' 'Pera,' or 'Seleta' orange trees [7]. The tree is late-ripening and moderately vigorous, with a medium-sized canopy and a rounded shape. The fruits have a slightly oval shape and medium size, skin with a slightly rough texture, a low number of seeds, and are difficult to harvest manually. The pulp is orange in color, with a high juice yield and low acidity. Fruits can be used in the fresh market and orange juice industry [8,9].

The cultivar Charmute de Brotas (*C. sinensis* [L.] Osbeck) has no known origin and may have been selected from the municipality of Engenheiro Coelho in São Paulo. This late-ripening cultivar is characterized by a high production of high-quality fruits, a small number of seeds, and a harvest period from October to April. It is a late-ripening cultivar [10].

Both cultivars have a late ripening period and a vocation to serve the table fruit market outside the off-season [11]. Late-ripening cultivars are important for citrus growth as they allow for better harvest distribution throughout the year. Because they produce fruits that remain on the tree for a long time without loss of quality, these cultivars allow the harvesting and commercialization of fruits in the off-season, a period when there is an increase in demand and a consequent rise in prices [5]. The cultivars Valencia and Folha Murcha represent almost all late oranges growing in Brazil and approximately 35% of orange production in the state of São Paulo [6].

Selecting proper rootstock can provide important improvements to the scion [12], such as juvenile period reduction, homogeneous tree architecture, pest and disease protection, water and nutrient absorption, tolerance to abiotic stress [13] and increased yield. Fruit size, juice quality, fruit ripening duration, sugar and acid content, fruit skin color, and fruit thickness are also influenced by rootstocks [14,15].

The 'Rangpur' lime (*Citrus limonia* Osbeck) tree is a natural hybrid of *Citrus medica* L. and mandarin (*Citrus reticulata* Blanco) and is suggested to be native to India [16]. In Brazil, the rootstock of 'Rangpur' lime has previously been used in citrus orchards due to its vigor, drought tolerance, high yield, precocity, and early fruit maturation [9]. Although it is tolerant to citrus tristeza virus (CTV), it is susceptible to citrus exocortis viroid (CEVd) and citrus sudden death-associated virus (SCDaV) [17].

'Swingle' [*Poncirus trifoliata* (L.) Raf; × *Citrus paradisi* Macf.] is the most cultivated citrumelo in Brazil and worldwide. It is one of the main rootstocks used to diversify orange groves and provides scions with high-quality fruits, high juice yield, high TSS content and yield, and low scion vigor. This cultivar is ideal for semi-dense planting in cooler locations and is not compatible with all scions [14]. It is resistant to citrus sudden death-associated viruses and decline [18].

Citrus production is strongly affected by environmental fluctuations that affect tree growth and development. Water scarcity and high temperatures are among the most common adverse climate changes worldwide that influence citrus growth and yield [12,19].

In the central-west region of the state of São Paulo, in the municipality of São Manuel, the hot season lasts 4.8 months, from November to April, with a maximum average daily temperature above 27 °C. The cold season lasts for 2.7 months, from May to August, with an average daily maximum temperature below 23 °C. The season with the heaviest precipitation lasts for 5 months, from October to March. January is the month with the highest precipitation. The dry season lasts for 7 months, from March to October. July is the month with the lowest precipitation [20].

Phenological studies of the scion, rootstock, and environmental interactions are important to identify the duration of the phenological stages of each scion and rootstock combination, aiming at staggering production during periods of lower fruit supply. In addition, it mitigates the risk of introducing new cultivars [15]. Studying the thermal demand by quantifying degree-days is an important tool for predicting phenological stages and harvesting [21,22].

The growing concern about the sustainability of citrus growth in Brazil indicates the need for studies assessing germplasm diversification in different growing regions that consider the great climatic diversity and aim at a better use of environmental resources. Therefore, in association with phenological studies, evaluating the thermal sum, vegetative and productive performance, fruit ripening curve, and harvest quality is essential. During ripening, the physical, physicochemical, and biochemical properties of fruits vary depending on the edaphoclimatic conditions of the growing region and the genotype of each cultivar [14]. Ripening is one of the first studies to be carried out in the characterization of new genotypes, as it enables not only the differentiation of cultivars but also the definition of the most appropriate harvest stage [23].

Considering the expansion of citrus orchards to new sites and the need for diversification of scion and rootstock combinations, the objective of this study was to evaluate the duration of phenological stages, thermal sum, vegetative and yield performance, alternate bearing, fruit ripening development, and quality of Folha Murcha (FM) and Charmute de Brotas (CB) orange cultivars on 'Rangpur' lime (RL) and 'Swingle' citrumelo (SC) rootstocks in the Midwest region of São Paulo State, southeastern Brazil.

## 2. Material and Methods

### 2.1. Experimental Site

The experiment was conducted at the São Manuel Experimental Farm, School of Agriculture, São Paulo State University (UNESP), Botucatu, São Paulo, Brazil (22°44′28″ S and 48°34′37″ W; 740 m above sea level). The climate of the region, according to the Köppen-Geiger classification, is *Cwa*, or warm temperate (mesothermal) and humid. The average temperature of the warmest month is above 22 °C, with an average annual precipitation of 1.377 mm. A weather station located 100 m from the experimental site recorded daily precipitation (mm) and maximum, minimum, and average temperatures (°C) throughout the experimental period; the water balance is shown in Figure 1. The water balance was determined by adopting an average root system depth of 80 cm and an available water capacity of 70 mm [24]. The soil is classified as sandy-textured Latossolo Vermelho distroférrico [25].

### 2.2. Plant Material and Crop Management

A replicated trial was conducted over two consecutive seasons, from August 2019 to June 2021. The orchard was planted in September 2016, with seedlings already grafted in the same year donated by the Coordination of Integral Technical Assistance, Department of Seeds, Seedlings, and Matrices. The trees were grown with 4 m spacing between trees and 6 m between rows, for a stand of 417 trees per hectare, without irrigation.

The experimental area was prepared based on the soil analysis and orange crop recommendations using plowing, sorting, and liming. Culture practices and fertilization were performed according to the standard management recommended for citrus orchards. 'Folha Murcha' (FM) and 'Charmute de Brotas' (CB) orange tree cultivars grafted onto 'Rangpur' lime (RL) and 'Swingle' citrumelo (SC) rootstocks were evaluated.

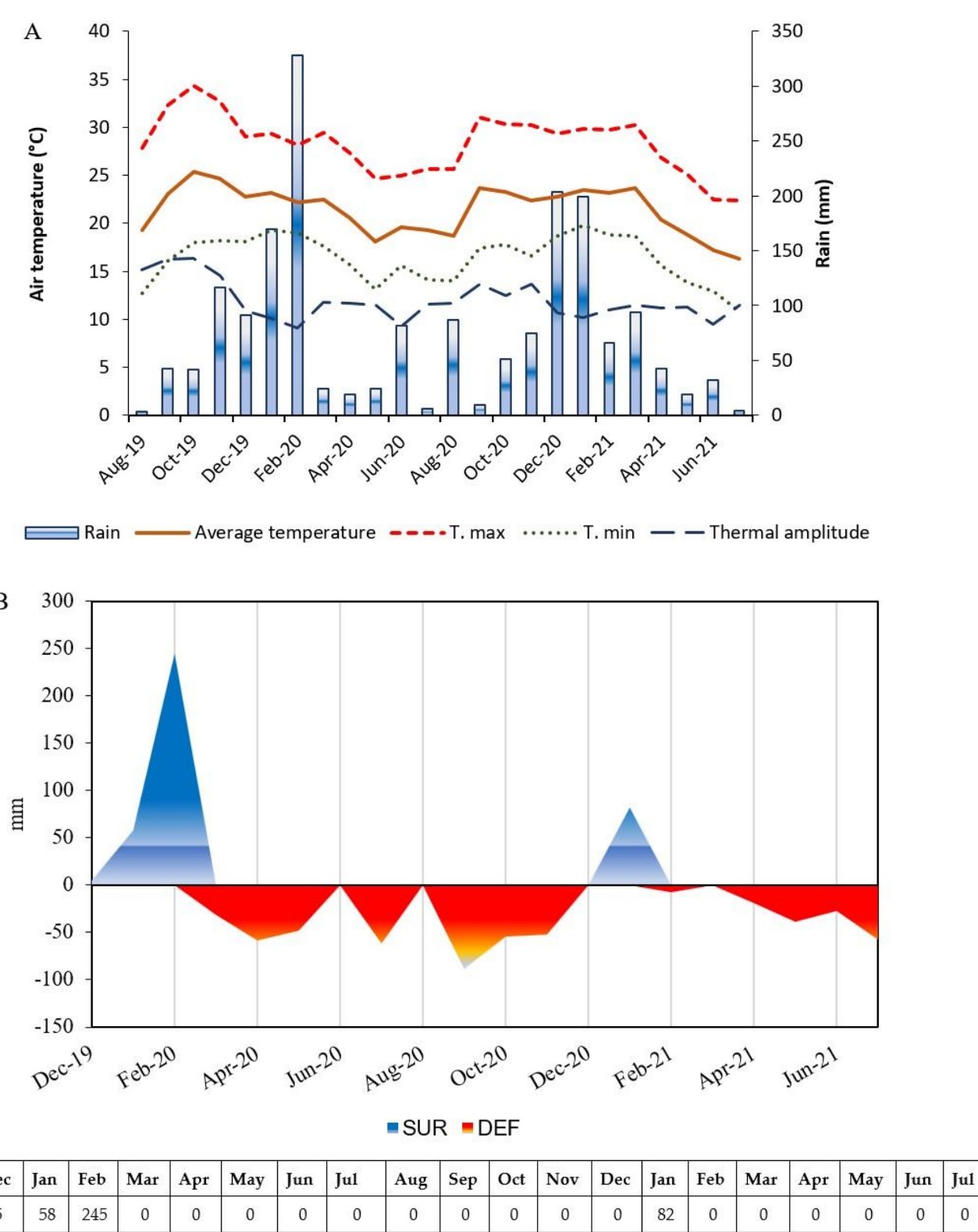

**Figure 1.** Precipitation, average, maximum, and minimum monthly temperatures, and thermal amplitude of the experimental period (**A**). Climatic water balance from August 2019 to July 2021: SUR = surplus, DEF = deficit (**B**).

|  | Dec | Jan | Feb | Mar | Apr | May | Jun | Jul | Aug | Sep | Oct | Nov | Dec | Jan | Feb | Mar | Apr | May | Jun | Jul |
|---|---|---|---|---|---|---|---|---|---|---|---|---|---|---|---|---|---|---|---|---|
| **SUR** | 5 | 58 | 245 | 0 | 0 | 0 | 0 | 0 | 0 | 0 | 0 | 0 | 0 | 82 | 0 | 0 | 0 | 0 | 0 | 0 |
| **DEF** | 0 | 0 | 0 | −31 | −58 | −48 | 0 | −62 | 0 | −89 | −54 | −52 | 0 | 0 | −7 | 0 | −19 | −39 | −27 | −57 |

### 2.3. Treatments and Experimental Design

The treatments consisted of four scion/rootstock combinations: 'Folha Murcha'/'Rangpur' lime (FM/RL), 'Folha Murcha'/'Swingle' citrumelo (FM/SC), 'Charmute de Brotas'/'Rangpur' lime (CB/RL), and 'Charmute de Brotas'/'Swingle' citrumelo (CB/SC). The experimental design was a randomized block in a split-plot arrangement (4 × 2). The plots were represented by four scion/rootstock combinations and subplots for two harvest seasons, with four replicates of three trees per plot, totaling 48 useful plants and guard trees external to the trial. A split-plot design, with the rootstocks as the plots and days of evaluation as the subplots, was used to evaluate fruit ripening.

### 2.4. Phenological Intervals and Thermal Sum

Each tree was divided into four quadrants, and in each quadrant, two randomly marked branches were evaluated based on the adapted BBCH (Biologische Bundesanstalt, Bundessortenamt, and Chemisch Industrie) scale [26,27] (Figure 2) for the phenology study. The trees were graded from 0 to 10, corresponding to the phenological stages, at 20-day intervals, starting when the fruits began to change color. Grades were assigned on each day and quadrant by branch evaluation, and the final grade for each tree corresponded to the average of the quadrants defined by the predominant phenological stage.

For the different sub-periods, the degree-day (DD) accumulation calculations were performed using Equations (1)–(3), according to the methodology proposed by [28]:

$$DD = \frac{TM + Tm}{2} - Tb \tag{1}$$

$$DD = \frac{(TM - Tb)^2}{2(TM - Tm)} \tag{2}$$

$$DD = 2\frac{(TM - Tm)(Tm - Tb) + (TM - Tm)^2 - (TM - TB)^2}{2(TM - Tm)} \tag{3}$$

Equation (1) was used when Tm > 12.8 °C and TM < 36 °C; Equation (2) when Tm < 12.8 °C and TM < 36 °C; and Equation (3) when Tm > 12.8 °C and TM > 36 °C, where Tm = minimum air temperature; TM = maximum air temperature; Tb = lower base temperature for citrus (12.8 °C); TB = upper-temperature limit (36 °C) [21,29,30].

### 2.5. Growth Performance

Growth performance was evaluated using the following variables, which were measured in September of each season:

(a). Tree height is measured in meters with a ruler graduated in centimeters.
(b). Stem diameter above and below the rootstock, measured in millimeters using a digital caliper. The insertion of the rootstock was used as a reference point, taking the diameter 10 cm above and 10 cm below this point.
(c). Canopy volume, determined according to Zekri [31]:

$$CAV = \left(\frac{\pi}{6}\right) \times TH \times L1 \times L2 \tag{4}$$

where CAV = canopy volume ($m^3$); TH = tree height (m); L1 = canopy width parallel to the planting line (m); L2 = crown width perpendicular to the planting line (m).

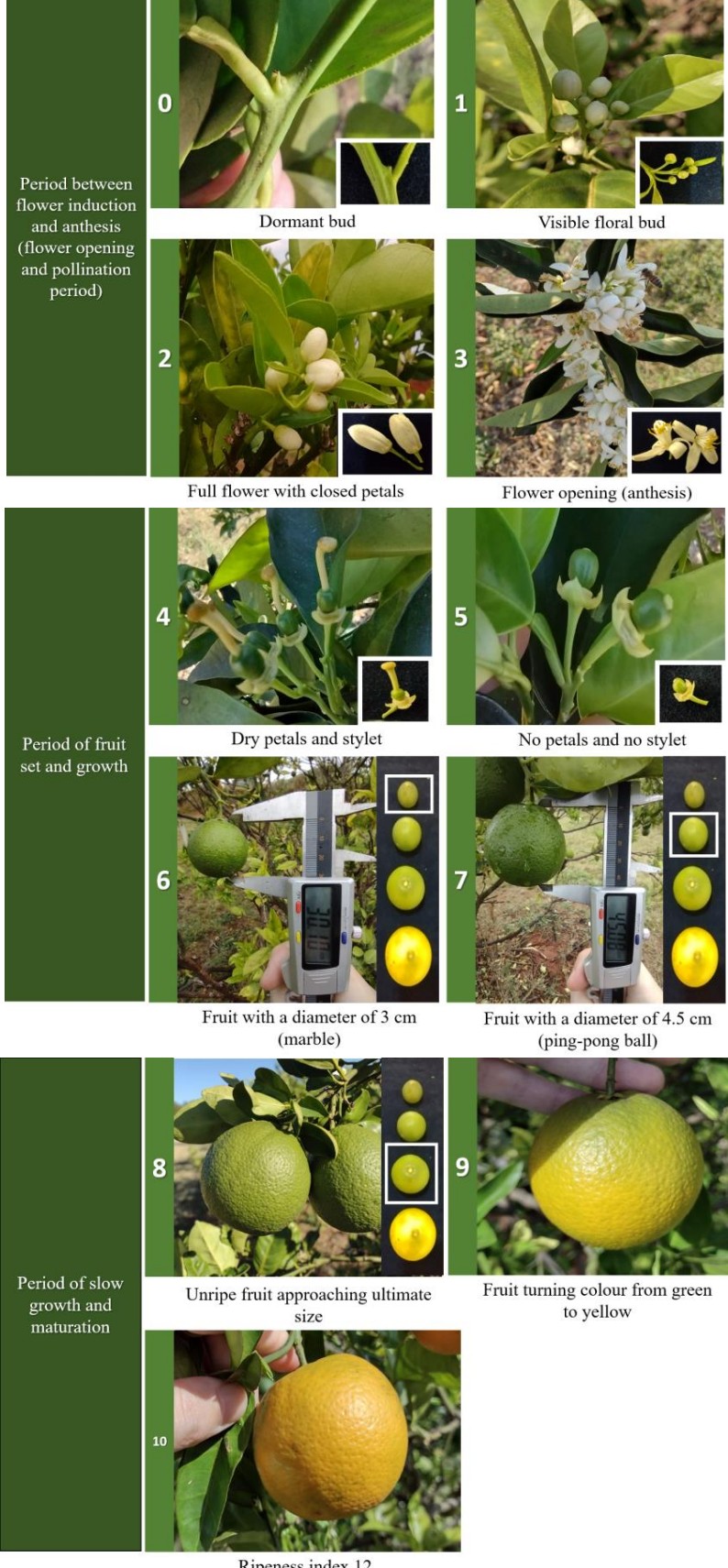

**Figure 2.** Score scale designed for the fruit growth and developmental stages of citrus trees. Source: Authors.

### 2.6. Yield Performance

The fruits of each treatment were harvested in October 2019 and 2020 for the CB cultivar and in January 2020 and 2021 for the FM cultivar when they reached a ripeness index of 16 for CB and from 10 to 13 for FM, which is considered appropriate for harvesting [8,14]. The variables analyzed were as follows:

(d). Yield (Y) (kg tree$^{-1}$) was quantified by counting and weighing fruits.

(e). Cumulative yield (CY) (kg tree$^{-1}$) obtained by adding the yields of the two harvests.

(f). Productive efficiency (PE) (kg m$^{-3}$): the relationship between CY and canopy volume (CAV) was used (Equation (5)) [32]:

$$PE = \frac{CY}{CAV} \tag{5}$$

(g). Productivity index: the relationship between the CY and the area of the tree stem section was calculated using Equation (6) [33]:

$$PI = \frac{CY}{SA} \tag{6}$$

where: PI = productivity index (kg cm$^{-2}$); CY = cumulative yield (kg tree$^{-1}$); SA = stem section area ($\pi R^2$). The average stem radius was calculated using Equation (7) [33]:

$$R = \frac{Pm}{2\pi} \tag{7}$$

where R is the stem radius, and Pm is the stem perimeter measured above and below the grafting.

(h). Alternate-bearing index (ABI) was calculated using Equation (8) [33].

$$ABI = 1/(n-1) \times \{|\alpha_2 - \alpha_1|/(\alpha_2 - \alpha_1) + |\alpha_3 - \alpha_2|/(\alpha_3 - \alpha_2) + \ldots + |\alpha_n + \alpha_{n-1}|/(\alpha_n + \alpha_{n-1})\} \tag{8}$$

where: ABI = alternate bearing index; n = number of years; and $\alpha_1$, $\alpha_2 \ldots \alpha_n$ are the productions of the corresponding years. The ABI can vary from 0 to 1, and the closer it is to zero, the lower the alternate bearing index between years.

### 2.7. Fruit Ripening Development and Quality

Fruit ripening development was evaluated using 10 fruits from each replicate, collected every 20 days, totaling 40 fruits per treatment. The fruits were collected from the outermost portion of the canopy of each plot at an average height of 1.5 m above the ground. The period in which the evaluations took place to establish ripening development was based on the time that the fruits of each cultivar were taken from the stage of fruit color change until they reached a ripeness index or ratio (total soluble solids/titratable acidity) of 16 for cultivar CB and from 10 to 13 for cultivar FM (Figure 3).

The variables evaluated during the fruit ripening period were:

(i). Juice yield (JY) (%): determined after measuring the juice mass and calculated by the relationship between juice mass/fruit mass.

(j). Fruit firmness (FF) (N): obtained with the aid of a texturometer (TA. XT Plus Texture Analyzer, Stable Micro Systems, Waverley, UK) with a SMS P/2 compression probe (Stable Micro Systems, Waverley, UK) using a speed of 1.0 mm s$^{-1}$ and compressing the fruit for 10 mm from the contact point. The reading was performed in the equatorial orientation of the fruit.

(k). Citrus color index (CCI) of fruit skin: obtained using a Minolta Chroma Meter CR-300 (Konica Minolta Sensing Americas Inc., Ramsey, NJ, USA) colorimeter measuring two equidistant points on the fruit. The values of a*, the variation between green and red color; b*, the variation between blue and yellow color; and L, the variation in

luminosity, from black (L = 0) to white (L-100). The values obtained were applied in Equation (9) to calculate the citrus color index of skin [34]:

$$CI = \frac{1000 \times a*}{(L \times b)} \tag{9}$$

(l).  Total soluble solids (TSS): were determined by adding three drops of the juice into a digital refractometer (Atago 3405 PR-32ª, Atago Co., Ltd., Tokyo, Japan), with automatic temperature compensation [35] (°Brix).

(m).  Titratable acidity (TA): measured by titrating 25 mL of juice with a standardized solution of sodium hydroxide (Merck KGaA, Darmstadt, Germany) at 0.1 N, using phenolphthalein (Merck KGaA, Darmstadt, Germany) as an indicator, which occurs when the potentiometer reaches 8.1 (% of citric acid) [36].

(n).  Ripeness index or Ratio (RI): determined through the relationship between total soluble solids and TA [37].

(o).  Ascorbic acid (AA): ascorbic acid content was determined according to the Association of Official Analytical Chemists method 967.21 [35]. The homogenized juice extract (10 mL) was diluted with 100 mL of 3% metaphosphoric acid (Merck KGaA, Darmstadt, Germany) and passed through filter paper. Next, 5 mL of the filtrate was titrated with 2,6-dichlorophenol iodophenol (DCPIP, Merck, Darmstadt, Germany) as the indicator. The results were expressed in mg of ascorbic acid per 100 g of fresh fruit weight. A standardization curve was created by titration with a standard solution containing a known amount of AA.

(p).  Technological index (TI) (kg TSS caixa$^{-1}$): was calculated using the equation proposed by [38] and expressed in kg TSS box$^{-1}$ (Equation (10)):

$$TI = \frac{(JY \times TSS \times 40.8)}{10{,}000} \tag{10}$$

where: JY = juice yield (%); TSS = total soluble solids (°Brix); 40.8 = standard weight of the box used to harvest oranges (kg).

During the main harvest of the two evaluated seasons, samples of 10 fruits per plot were collected, totaling 80 fruits per treatment, for the evaluation of the following variables:

(q).  Fruit weight (PF) was obtained by weighing the fruit (g).

(r).  Juice weight (JW): Juice was extracted using a semi-industrial juicer and weighed (g).

(s).  Fruit length (FL) and diameter (DF): The length and diameter of the fruits were measured using a digital caliper (mm).

(t).  Number of seeds per fruit (NS): seeds were extracted and counted, and the average between the fruits of each cultivar was calculated.

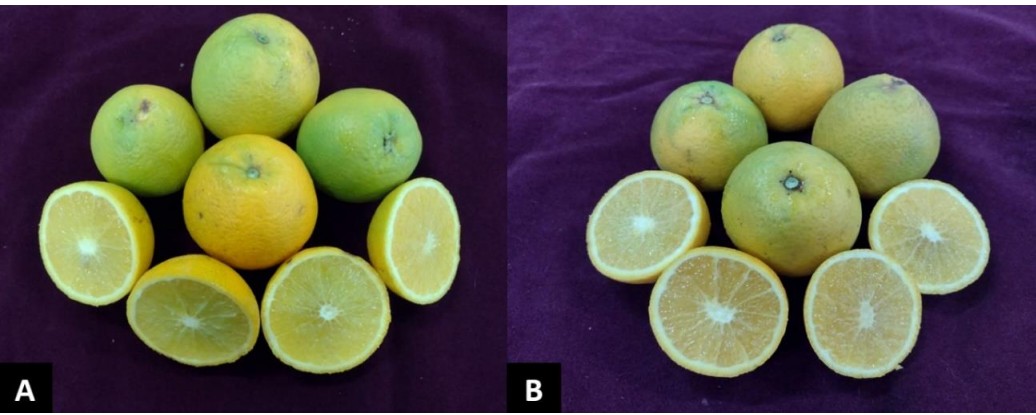

**Figure 3.** 'Folha Murcha' (**A**) and 'Charmute de Brotas' (**B**) grafted onto 'Swingle' citrumelo orange fruits. Source: Authors.

### 2.8. Statistical Analysis

Two-year data on the evaluated variables were analyzed as repeated measures, and the reported data are the means of the two years of evaluation.

Data were subjected to analysis of variance, and when there was significance at 1% and 5% probability and when the F test established significance, means were compared using Tukey's test. For the evaluation of fruit ripening development, the rootstocks were also compared using Tukey's test, and the days of evaluation were compared using regression analysis. AgroEstat data analysis software was used for all variables.

## 3. Results

### 3.1. Harvesting Seasons

There was a statistical difference between the two harvesting seasons evaluated for the following variables: SDBG, SDAG, TH, Y, PE, PI, FL, FD (Table 1), and FF for the FM/RL and FM/SC combinations, and JY and CCI for the CB/RL and CB/SC combinations (Table 2).

**Table 1.** Analysis of variance of 2019–2020 and 2020–2021 harvest seasons of stem diameter below grafting (SDBG), stem diameter above grafting (SDAG), tree height (TH), canopy volume (CAV), yield (Y), production efficiency (PE), productivity index (PI), number of seeds (NS), fruit length (FL), and fruit diameter (FD) of the scion/rootstock combinations.

|  | Scion/Rootstock | |
| --- | --- | --- |
|  | **F-Value** | **CV (%)** |
| SDBG | 54.96 ** | 0.92 |
| SDAG | 7.92 * | 7.95 |
| TH | 62.74 ** | 0.85 |
| CAV | 0.26 ns | 18.06 |
| Y | 113.54 ** | 20.69 |
| PE | 74.36 ** | 12.31 |
| PI | 102.90 ** | 21.82 |
| NS | 0.11 ns | 23.24 |
| FL | 20.07 ** | 6 |
| FD | 19.76 ** | 6.88 |

** = statistically different at 1%; * = statistically different at 5%; ns = do not differ statistically by the F-test. CV = coefficient of variation.

**Table 2.** Analysis of variance of 2019–2020 and 2020–2021 harvest seasons of juice yield (JY), fruit firmness (FF), citrus color index (CCI), total soluble solids (TSS), titratable acidity (TA), ripeness index (RI), ascorbic acid (AA), and technological index (TI) of the FM/RL and FM/SC; CB/RL and CB/SC combinations.

|  | FM/RL and FM/SC | | CB/RL and CB/SC | |
| --- | --- | --- | --- | --- |
|  | **F-Value** | **CV (%)** | **F-Value** | **CV (%)** |
| JY | 0.28 ns | 1.68 | 29.72 ** | 5.92 |
| FF | 564.32 ** | 0.8 | 5.33 ns | 0.76 |
| CCI | 0.19 ns | 19.32 | 51.86 ** | 2.64 |
| TSS | 2.26 ns | 3.54 | 0.02 ns | 1.94 |
| TA | 1.09 ns | 5.49 | 0.24 ns | 4.86 |
| RI | 3.25 ns | 6.02 | 0.07 ns | 5.64 |
| AA | 0.01 ns | 2.75 | 3.12 ns | 8.89 |
| TI | 0.18 ns | 6.08 | 0.12 ns | 1.2 |

** = statistically different at 1%; ns = do not differ statistically by the F-test. CV = coefficient of variation; FM = 'Folha Murcha'; CB = 'Charmute de Brotas'; RL = 'Rangpur' lime; SC = 'Swingle' citrumelo.

### 3.2. Phenological Intervals and Thermal Sum

There was no significant interaction between scion and rootstock cultivars for cycle length (scores from 0 to 9), measured in days and degree-days. However, there was a signif-

icant difference between the rootstocks. Trees from the FM/SC and CB/SC combinations had longer cycle lengths than the trees grafted onto RL (Table 3).

There was no difference in the duration of the phenological interval between scion/rootstock combinations. It was observed that, on average, the duration of subperiod 0–1 was 40 days, accumulating 294.33 DD, whereas the trees remained in subperiod 1–3 for 23 days, accumulating 176 DD, regardless of the combination. Therefore, the interval between the flower bud stage and anthesis (0–3) was 63 days and accumulated at 470.33 DD (Figure 4A,B).

**Table 3.** Phenological cycle of the FM/RL, FM/SC, CB/RL and CB/SC scion/rootstock combinations is measured in days and degree-days.

| Rootstock | Days (0–9) [1] | Degree-Days (0–9) |
|---|---|---|
| 'Rangpur' lime | 390.44 b | 3522.56 b |
| 'Swingle' citrumelo | 421.63 a | 3689.62 a |
| MSD | 0.07 | 0.02 |
| CV (%) | 1.07 | 0.23 |
| F-value | 5.37 * | 20.45 ** |

** = statistically different at 1%; * = statistically different at 5%. MSD = Minimum significant difference; CV = coefficient of variation. Means followed by different letter in the column differ from each other by the Tukey test at 1 or 5% probability. [1] (0) dormant bud; (1) visible floral bud; (2) full flower with closed petals; (3) flower opening (anthesis); (4) dry petals and stylet; (5) no petals and no stylet; (6) fruit with a diameter of 3 cm (marble); (7) fruit with diameter of 4.5 cm (ping-pong ball); (8) unripe fruit approaching ultimate size; (9) fruit turning color from green to yellow.

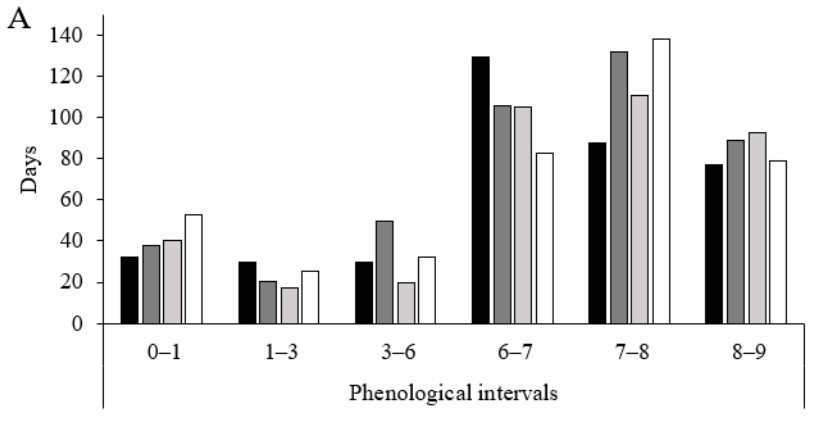

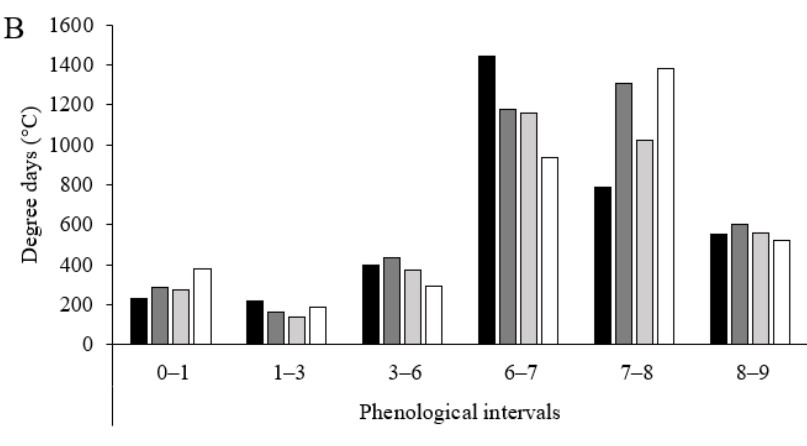

**Figure 4.** Phenological intervals of the scion/rootstock combinations measured in days (**A**) and degree-days (**B**). FM = 'Folha Murcha'; CB = 'Charmute de Brotas'; RL: 'Rangpur' lime; SC: 'Swingle' citrumelo.

The variables evaluated showed that the combinations did not affect the time of flower induction because this occurred in August for all scions/rootstocks.

The period between anthesis and the start of fruit ripening (subperiods 3–9) was 377 days and accumulated 3238.5 DD. The longest sub-periods were 6–7 and 7–8, with durations of 105 and 116 days and accumulating 1180 and 1126.14 DD, respectively, which comprise the exponential and linear growth periods of the fruits [27].

### 3.3. Growth Performance

The variables related to growth performance showed a difference between crop seasons (Table 1), with higher means in the second harvesting season evaluated due to a small growth observed in the trees depending on their age (3 and 4 years, respectively).

There was a significant interaction between scions and rootstocks for TH. For CAV, significant differences were observed between the scion and rootstock cultivars. The stem diameter above the grafting point (SDAG) differed only between the rootstocks (Table 4). CB/RL had a lower TH than FM/RL. However, CB/SC exhibited a higher TH than FM/SC.

RL produced greater CAV and SDAG than did SC (Table 4). Comparing the CAV between the scion cultivars, FM had lower CAV values than CB.

**Table 4.** Analysis of variance (ANOVA) and means comparison of stem diameter below grafting (SDBG), stem diameter above grafting (SDAG), tree height (TH), and canopy volume (CAV) of the different scion/rootstock combinations.

| | DF | SDBG | SDAG | TH | CAV |
|---|---|---|---|---|---|
| S (A) | 1 | 4.65 ns | 0.93 ns | 3.41 ns | 11.25 ** |
| R (B) | 1 | 0.26 ns | 8.97 * | 35.94 ** | 11.16 ** |
| Block | 3 | 3.45 ns | 0.19 ns | 1.00 ns | 0.32 ns |
| S × R | 1 | 0.99 ns | 0.28 ns | 14.75 ** | 0.45 ns |
| CV (%) | | 33.60 | 3.22 | 16.81 | 19.85 |
| **Rootstock** | **CAV (m$^3$) $^1$** | **SDAG (mm)** | **Scions** | **CAV (m$^3$)** | |
| RL | 7.80 a | 89.43 a | FM | 5.28 b | |
| SC | 5.29 b | 72.61 b | CB | 7.81 a | |
| MSD | 0.66 | 0.15 | DMS | 0.66 | |
| **Tree height (m) $^2$** | | | | | |
| **Rootstock** | | | | | |
| Scion | RL | SC | | | |
| FM | 2.75 Aa | 2.03 Ab | | | |
| CB | 2.37 Ba | 2.20 Aa | | | |
| MSD | 0.79 | | | | |

** = statistically different at 1%; * = statistically different at 5%; ns = do not differ statistically by the F test, $p < 0.05$. S = Scion; R = Rootstock. $^1$ Means followed by different letters in the column differ statistically by Tukey test at 5% probability level. $^2$ Means followed by different letter, upper case letter in the column and lower-case letter in the row; differ statistically by Tukey test at 5% probability level. DF = Degree of freedom; CV = Coefficient of variation; MSD = Minimum significant difference; FM = 'Folha Murcha'; CB = 'Charmute de Brotas'; RL = 'Rangpur' lime; SC = 'Swingle' citrumelo.

### 3.4. Yield Performance

The variables of yield performance as well as the diameter and length of the fruits differed between harvesting seasons (Table 1), since the production of the second harvest was significantly lower than that of the first. This difference may be associated with the drought that occurred in the second crop evaluated (Figure 1).

There was a significant interaction between scion and rootstock cultivars in the fruit yield harvested in the 2019/2020 season. However, in the 2020/2021 season, only scions showed a significant difference (Table 5).

The average yield of the combinations during the 2019/2020 season varied from 23.89 to 14.35 kg tree$^{-1}$, with CB/SC being the most productive and FM/SC the least productive (Figure 5). In the 2020/2021 crop year, there was an isolated effect of scion cultivars, with

CB showing higher production than FM, with average yields of 12.47 and 5.12 kg tree$^{-1}$, respectively (Figure 5).

**Table 5.** Analysis of variance of scion (S), rootstock (R) and their interaction of yield (Y) of harvest from 2019–2020 and 2020/2021, cumulative yield (CY), production efficiency (PE), productivity index (PI), and alternate bearing index (ABI) of the different scion/rootstock combinations.

| | DF | Yield (kg tree$^{-1}$) | |
|---|---|---|---|
| | | **2019/2020** | **2020/2021** |
| S (A) | 1 | 5.44 * | 33.22 ** |
| R (B) | 1 | 1.99 $^{ns}$ | 0.88 $^{ns}$ |
| Block | 3 | 2.05 $^{ns}$ | 0.38 $^{ns}$ |
| S × R | 1 | 10.28 * | 1.40 $^{ns}$ |
| CV (%) | | 1.91 | 6.33 |

| | GL | Y | CY | PE | ABI |
|---|---|---|---|---|---|
| S (A) | 1 | 9.38 ** | 0.22 $^{ns}$ | 2.03 $^{ns}$ | 48.85 ** |
| R (B) | 1 | 1.29 $^{ns}$ | 5.53 * | 0.62 $^{ns}$ | 10.63 ** |
| Block | 3 | 1.45 $^{ns}$ | 0.50 $^{ns}$ | 2.95 $^{ns}$ | 2.93 $^{ns}$ |
| S × R | 1 | 10.58 * | 1.51 $^{ns}$ | 4.00 $^{ns}$ | 2.31 $^{ns}$ |
| CV (%) | | 1.14 | 34.14 | 17.00 | 8.70 |

** = statistically different at 1%; * = statistically different at 5%; $^{ns}$ = do not differ statistically by the F test $p < 0.05$. DF = Degree of freedom; CV = Coefficient of variation.

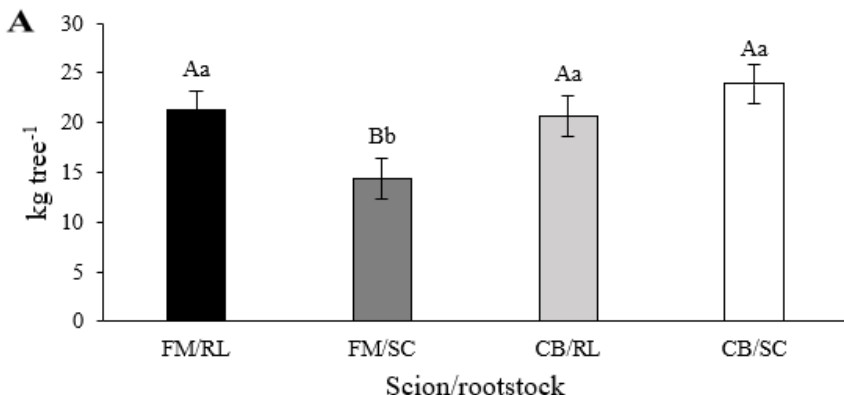

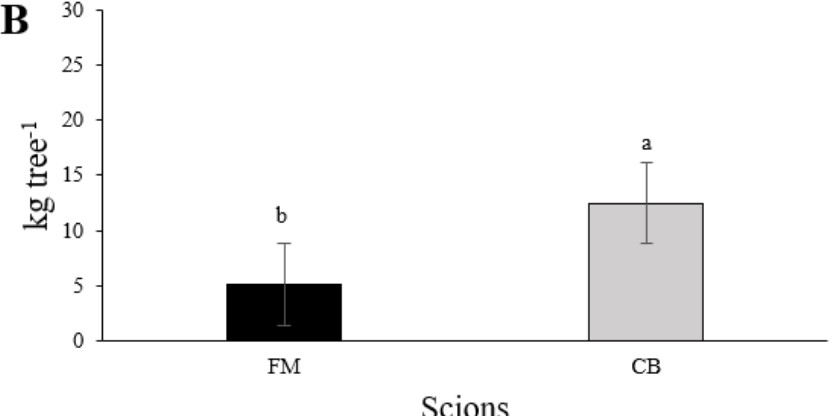

**Figure 5.** Yield (kg tree$^{-1}$) from 2019–2020 harvest of the rootstock/scion combinations (**A**) and from 2020–2021 of the sweet orange cultivars (**B**). FM = 'Folha Murcha'; CB = 'Charmute de Brotas'; RL = 'Rangpur' lime; SC = 'Swingle' citrumelo. Means followed by different letter differ from each other by the Tukey test at 5% probability.

There was a significant interaction between scion and rootstock for CY. In PE, there was a difference between rootstocks, and for ABI, there was a difference between scions and rootstocks. The PI showed no difference between treatments, with an average value of 3.44 kg cm$^{-2}$ for scion/rootstock combinations (Table 6).

FM/SC showed inferior performance to the other treatments for CY (Table 6). FM/SC exhibited lower CY values than CB/SC.

The ABI was higher for the RL than for the SC. Between the scions, FM had a higher ABI than CB (Table 6). The SC presented higher PE than the RL, as it resulted in lower vegetative vigor in the grafted scions.

**Table 6.** Cumulative yield (CY) of the different scion/rootstock combinations; alternate bearing index (ABI); and productive efficiency (PE) of the sweet orange cultivars and rootstocks.

| CY (kg tree$^{-1}$) [1] | | |
|---|---|---|
| | **Rootstock** | |
| **Scion** | **RL** | **SC** |
| FM | 22.97 Aa | 16.02 Bb |
| CB | 24.02 Aa | 28.91 Aa |
| MSD | 0.02 | |

| **Rootstock** | **ABI [2]** | **PE (kg m$^3$) [2]** | **Scion** | **ABI [2]** |
|---|---|---|---|---|
| RL | 0.48 a | 3.08 b | FM | 0.54 a |
| SC | 0.35 b | 4.26 a | CB | 0.29 b |
| MSD | 0.02 | 0.89 | MSD | 0.02 |

[1] Means followed by different letters, upper case letter in the column and lower-case letter in the row, differ statistically by Tukey test at 5% probability level. [2] Means followed by different letters in the column differ statistically by Tukey test at 5% probability level. MSD = Minimum significant difference; FM = 'Folha Murcha'; CB = 'Charmute de Brotas'; RL = 'Rangpur' lime; SC = 'Swingle' citrumelo.

### 3.5. Fruit Ripening Development and Quality

There was a significant effect of days after anthesis (DAA) on CCI ($p < 0.01$) for FM, with no difference between rootstocks. The two combinations showed a quadratic relationship for CCI, reaching a maximum value (2.87) at 94 DAA (Figure 6A). In general, the CCI was low during the ripening of FM fruits, reaching an average value close to zero at 180 DAA (Figure 4A), as the fruits showed a greenish skin color throughout the period [39].

CB had a significant effect on the CCI between rootstocks ($p < 0.05$) and DAA ($p < 0.05$). Both combinations showed a linear increase in CCI as a function of DAA (Figure 7A). The CB + SC combination produced fruits with a higher CCI than the CB + RL combination (Table 7).

At harvest, the fruits of the CB orange trees had higher CCI values than those of the FM trees. The fruits of CB had a skin color close to intense orange, whereas those of FM were close to light green.

There was no significant interaction for FF between scion/rootstock and DAA combinations. For both canopy cultivars, there was a significant effect of DAPA ($p < 0.01$), with a linear reduction for all combinations (Figures 6B and 7B).

The JY of the FM showed a significant interaction between the scion/rootstock and DAA ($p < 0.01$). A quadratic model was used to fit the JY for both combinations (Figures 6C and 7C). The FM/RL reached the maximum value of JY (46.64%) at 63 DAA, while the FM/SC reached the maximum value (54.34%) at 94 DAA. Between 60 and 140 DAA, FM/SC produced fruits with higher JY averages than those of RL. At 160 and 180 DAA, the two combinations showed similar JY values. Regarding CB, the JY was significantly affected only by DAA ($p < 0.05$). There was a quadratic increase in JY as a function of DAA, reaching the maximum value (47.67%) at 56 DAA.

**Table 7.** Citrus color index (CCI), total soluble solids (TSS), titratable acidity (TA), and technological index (TI) in fruits from CB/RL and CB/SC combinations, at different harvest times.

| Rootstock | CCI | TSS (°Brix) | TA (% Citric Acid) |
|---|---|---|---|
| RL | 1.75 b | 9.24 b | 0.85 b |
| SC | 2.68 a | 11.00 a | 1.00 a |
| MSD | 1.75 | 0.62 | 0.13 |
| CV (%) | 28.29 | 11.13 | 12.36 |
| F-value | 12.23 * | 15.46 * | 18.07 * |
| **Rootstock** | **TI (kg TSS box$^{-1}$)** | | |
| | FM | | CB |
| RL | 1.46 b | | 1.46 b |
| SC | 2.06 a | | 2.13 a |
| MSD | 0.20 | | 0.74 |
| CV (%) | 32.15 | | 23.61 |
| F-value | 30.38 * | | 38.05 ** |

** = statistically different at 1%; * = statistically different at 5%. Means followed by different letters in the column differ statistically by Tukey test at 5% probability level. MSD = Minimum significant difference; CV = coefficient of variation; FM = 'Folha Murcha'; CB = 'Charmute de Brotas'; RL = 'Rangpur' lime; SC = 'Swingle' citrumelo.

A comparison between the canopies showed that CB produced fruits with a higher JY than that of FM. For both scions, the JY ranged from 40 to 55% between the first and last evaluations, above the 35% required by the juice processing industries.

DAA had a significant effect on the TSS content in the scion cultivars (Figures 6D and 7D). However, for CB, there was a difference in TSS content between the rootstocks. During fruit ripening development, the fruits produced in SC had an average TSS content of 11 °Brix, while in RL, they had an average of 9.24 °Brix.

In CB, TSS increased linearly as a function of DAA, reaching a value of 10.93 °Brix at 180 DAA (Figure 7D). The FM orange fruits showed a quadratic increase in TSS as a function of DAA, reaching a maximum value (9.25 °Brix) at 99 DAA (Figure 6D).

The SC rootstock showed a higher TSS than RL, whereas FM produced fruits with a TSS content lower than that of CB. The fruits of the CB orange trees had an average RI value or ratio higher than that of the FM orange trees.

In the fruits produced by FM, there was a significant interaction between the TA content of the scion/rootstock combinations and DAA ($p < 0.01$) (Figure 6E). The TA of FM/RL exhibited a quadratic reduction, whereas that of FM/SC decreased linearly during fruit ripening. At 180 DAPA, FM/SC fruits showed lower TA than FM/RL fruits (Figure 6E).

In CB, there was no significant interaction between rootstock and DAA for TA (Figure 7E). There was an effect of rootstock ($p < 0.05$) and DAA ($p < 0.01$) on TA (Table 7). During fruit maturation, there was a linear decrease in TA as a function of DAA content in both rootstocks. The SC rootstock produced fruits with higher levels of citric acid than the plants grafted onto RL, with averages of 1.00 and 0.85% of citric acid, respectively.

There was a significant interaction between scion/rootstock and DAA ($p < 0.05$) for the RI of FM orange trees, with linear progression as a function of DAA for both rootstocks. The increase in RI, or ratio, for the two rootstocks was similar up to 60 DAA; however, between 80 and 180 DAA, the fruits produced by FM/SC showed a more accentuated increase than those produced by FM/RL (Figure 6F). DAA significantly influenced the RI ($p < 0.01$) of fruits produced by CB trees during ripening. Positive linear growth as a function of DAA was observed for both rootstocks (Figure 7F).

The FM/SC fruits reached the optimum harvest stage at 180 DAA, whereas the CB fruits reached the optimum harvest stage at 80 DAA.

The AA content of the FM cultivar showed a significant interaction between scion/ rootstock and DAA ($p < 0.05$). AA content decreased linearly during the maturation of fruits produced by FM/RL, whereas a linear model was used to adjust AA values as a function of DAA for fruits produced by trees grafted onto SC (Figure 6G). The AA content

decreased similarly during fruit ripening; however, at 180 DAA, fruits produced with FM/RL showed higher AA than those produced with FM/SC. The AA in the fruits of the CB cultivar had a significant effect on DAA ($p < 0.01$), with a quadratic reduction observed after 18 DAA (Figure 7G).

There was no significant interaction between scion/rootstock and DAA for the TI. However, rootstock and DAA had isolated effects. SC produced fruits with a TI higher than that of RL in both scion cultivars (Table 7).

CB/SC presented the highest TI, differing in relation to scions and rootstocks only from FM/RL, which presented the lowest TI. The TI of FM fruits responded to a quadratic regression model as a function of DAA, reaching a maximum content (1.86 kg TSS box$^{-1}$) at 87 DAA (Figure 6H), whereas the CB fruits showed linear growth (Figure 7H).

There were no significant interactions between the scion/rootstock combinations and the physical characteristics of the fruits. However, FL differed between the scions and rootstocks. There was a difference between the scions for the NS, while the DF differed between the rootstocks (Table 8).

CB had a lower NS per fruit and a higher FL than FM orange fruits; however, they did not differ in DF. The fruits obtained from the trees grafted on the RL rootstock had greater lengths and diameters than those of the trees grafted on the SC rootstock (Table 8).

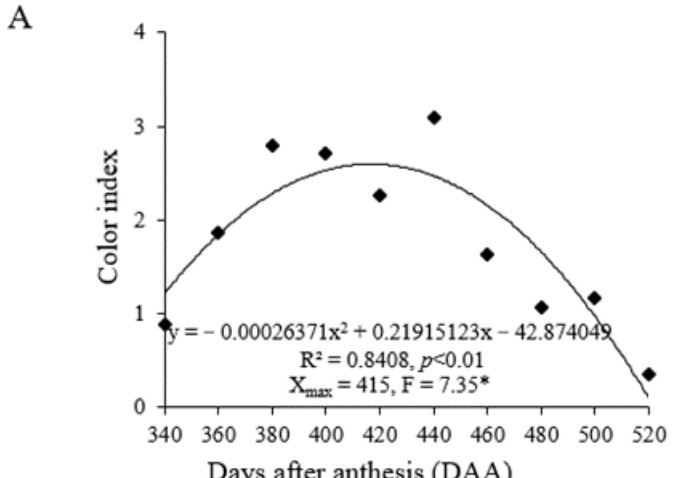

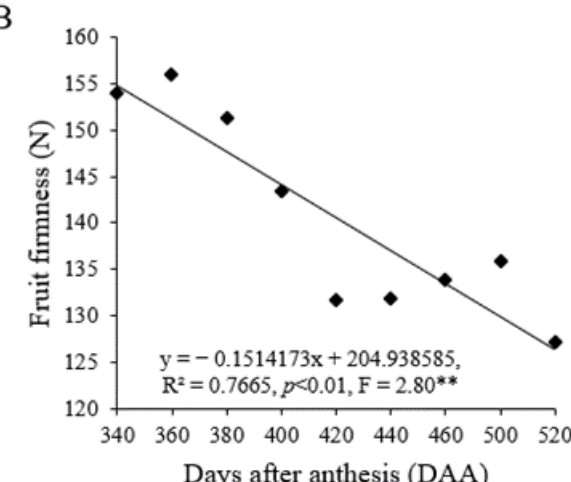

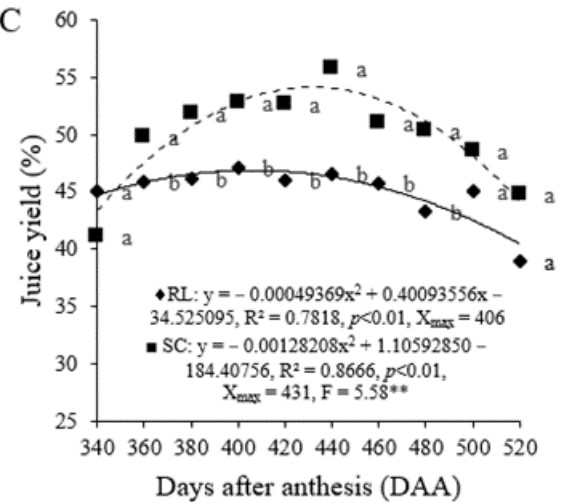

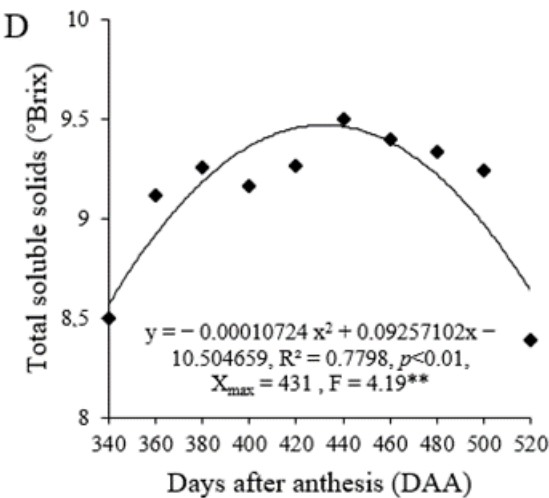

**Figure 6.** *Cont.*

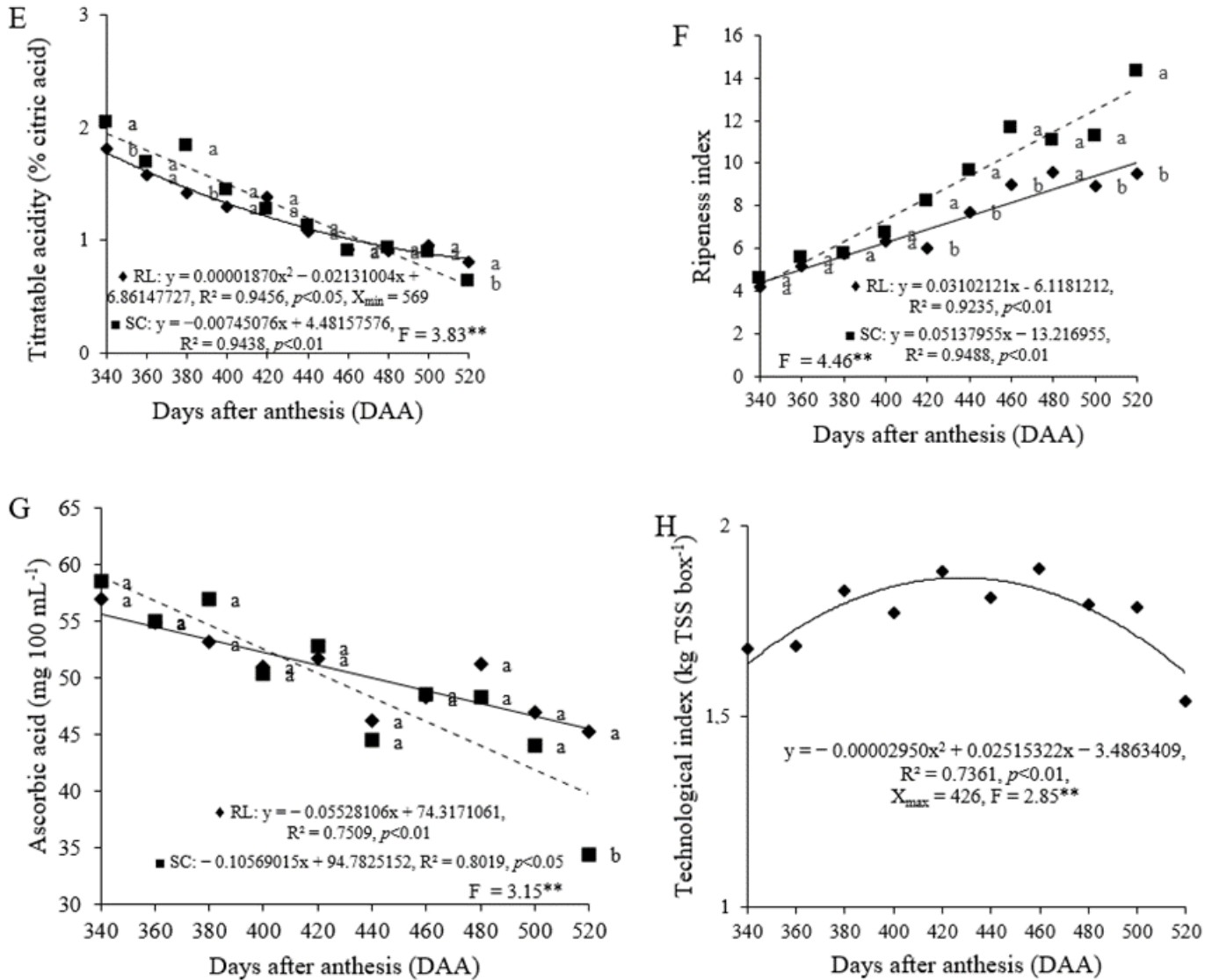

**Figure 6.** Citrus color index (**A**), fruit firmness (**B**), juice yield (**C**), total soluble solids (**D**), titratable acidity (**E**), ripeness index (**F**), ascorbic acid (**G**), and technological index (**H**) of the FM/RL and FM/SC combinations, at different harvest times. FCA/UNESP, São Manuel, SP, 2021. ** = statistically different at 1%; * = statistically different at 5%. Means followed by different letter differ from each other by the Tukey test at 1 or 5% probability. FM = 'Folha Murcha'; RL = 'Rangpur' lime; SC = 'Swingle' citrumelo.

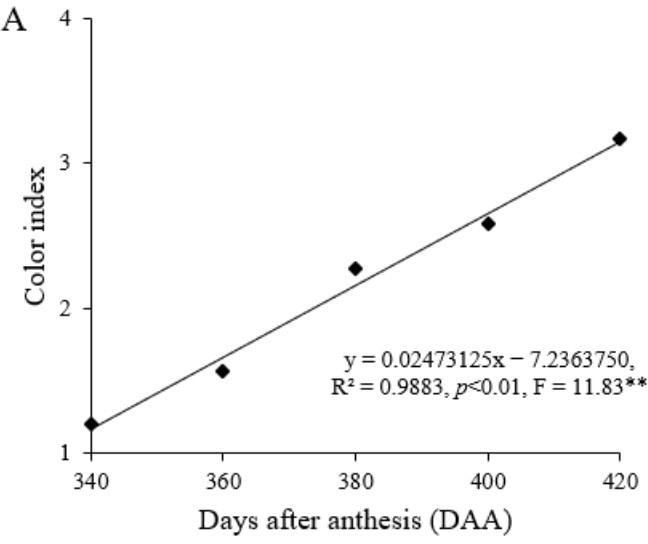

$y = 0.02473125x - 7.2363750$,
$R^2 = 0.9883$, $p<0.01$, $F = 11.83**$

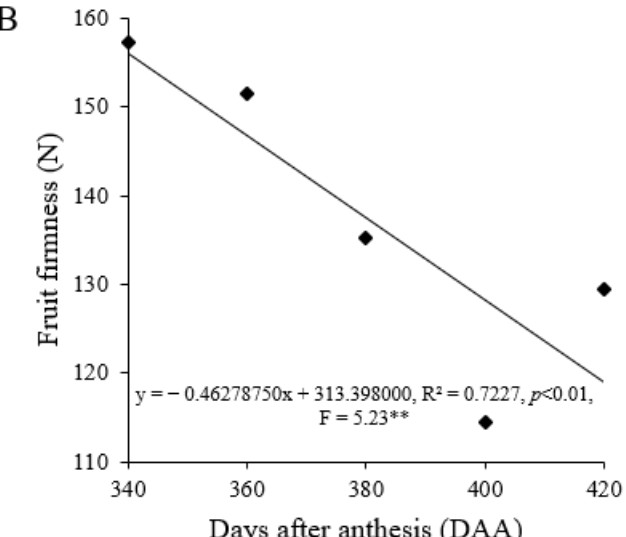

$y = -0.46278750x + 313.398000$, $R^2 = 0.7227$, $p<0.01$,
$F = 5.23**$

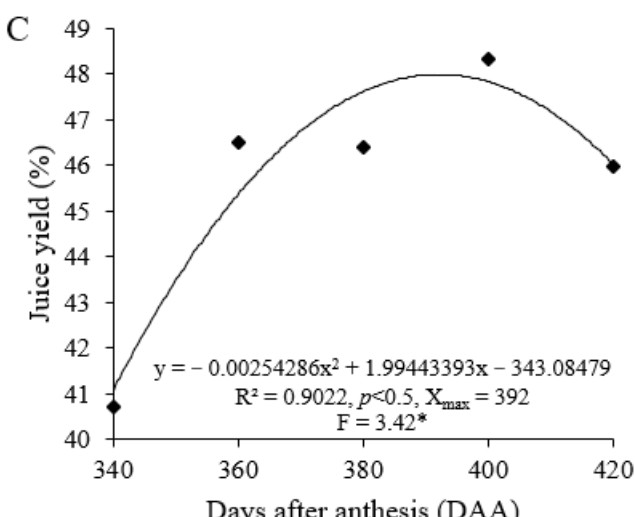

$y = -0.00254286x^2 + 1.99443393x - 343.08479$
$R^2 = 0.9022$, $p<0.5$, $X_{max} = 392$
$F = 3.42*$

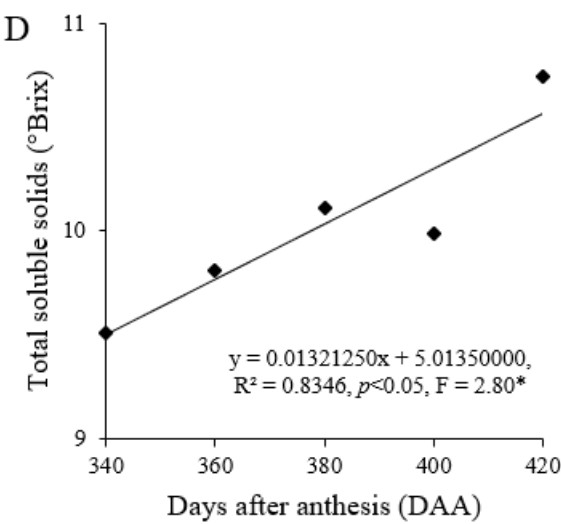

$y = 0.01321250x + 5.01350000$,
$R^2 = 0.8346$, $p<0.05$, $F = 2.80*$

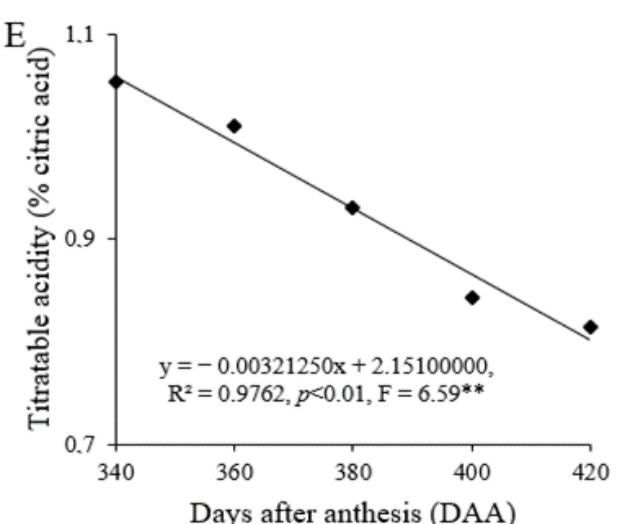

$y = -0.00321250x + 2.15100000$,
$R^2 = 0.9762$, $p<0.01$, $F = 6.59**$

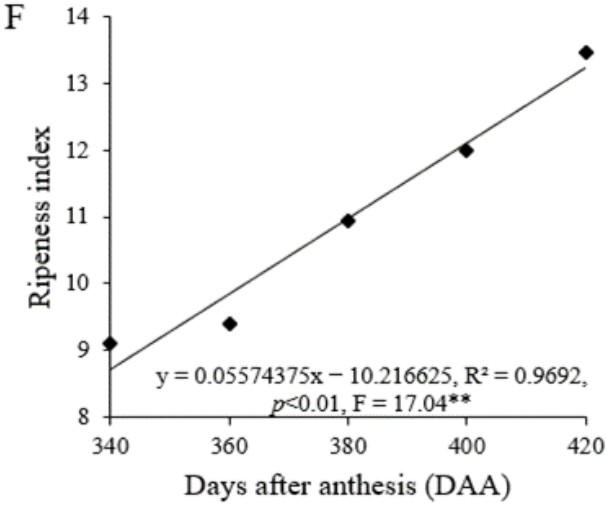

$y = 0.05574375x - 10.216625$, $R^2 = 0.9692$,
$p<0.01$, $F = 17.04**$

**Figure 7.** *Cont.*

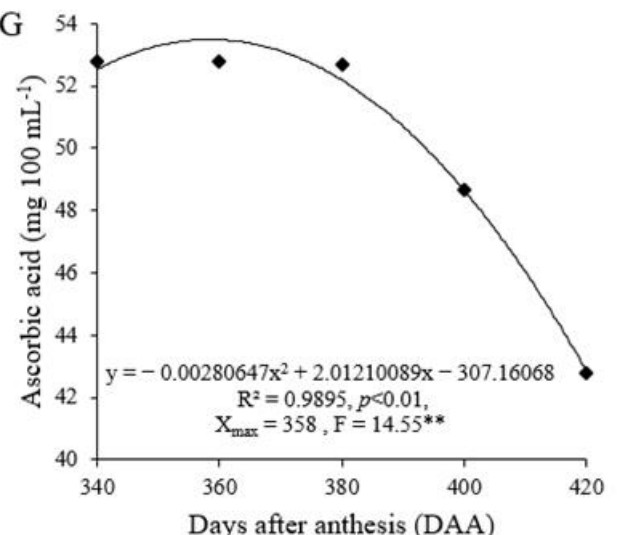
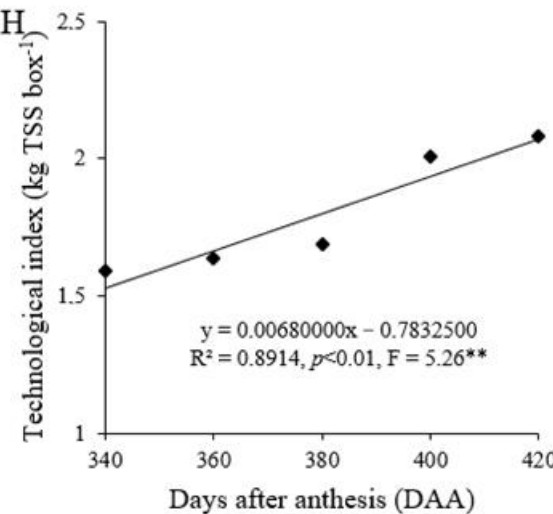

**Figure 7.** Citrus color index (**A**), fruit firmness (**B**), juice yield (**C**), total soluble solids (**D**), titratable acidity (**E**), ripeness index (**F**), ascorbic acid (**G**), and technological index (**H**) of the CB/RL and CB/SC combinations, at different harvest times. FCA/UNESP, São Manuel, SP, 2021. ** = statistically different at 1%; * = statistically different at 5%. CB = 'Charmute de Brotas'; RL = 'Rangpur' lime; SC = 'Swingle' citrumelo.

**Table 8.** Analysis of variance (ANOVA) and means comparison of number of seeds (NS), fruit length (FL), and fruit diameter (FD) of the scion/rootstock combinations.

|  | DF | NS | FL | FD |
|---|---|---|---|---|
| S (A) | 1 | 30.77 * | 6.76 * | 0.46 $^{ns}$ |
| R (B) | 1 | 7.12 $^{ns}$ | 45.42 ** | 114.17 ** |
| Block | 3 | 1.24 $^{ns}$ | 1.40 $^{ns}$ | 0.16 $^{ns}$ |
| S × R | 1 | 2.10 $^{ns}$ | 0.34 $^{ns}$ | 0.32 $^{ns}$ |
| CV (%) |  | 8.77 | 0.74 | 0.46 |
| **Rootstock** |  | **NS** | **FL (mm)** | **FD (mm)** |
| RL |  | - | 76.70 a | 77.51 a |
| SC |  | - | 71.52 b | 69.64 b |
| MSD |  | - | 0.13 | 0.02 |
| Scion |  |  |  |  |
| FM |  | 9.12 a | 72.75 b |  |
| CB |  | 4.42 b | 75.47 a |  |
| MSD |  | 0.60 | 0.13 |  |

** = statistically different at 1%; * = statistically different at 5%; $^{ns}$ = do not differ statistically by the F test $p < 0.05$. S = Scion; R = Rootstock; CV = coefficient of variation; DF = Degree of freedom. Means followed by different letters in the column differ statistically by Tukey test at 5% probability level. MSD = Minimum significant difference; FM = 'Folha Murcha'; CB = 'Charmute de Brotas'; RL = 'Rangpur' lime; SC = 'Swingle' citrumelo.

## 4. Discussion

The climate in the study region was atypical during the study period. Differences in precipitation patterns were observed during the two evaluation years (Figure 1A). Temperatures varied as expected for the region during the study period; however, in 2019, more days and hours of minimum temperatures occurred during winter, and the temperature range was greater than expected. The flowering and fruiting periods showed greater variation in spring 2019, owing to environmental conditions and scion/rootstock combinations, with an intense fruit drop. The year 2020 was characterized by an uneven rainfall distribution, with a dry period occurring at the time of the new flushing. The accumulated water deficit (394 mm) mainly affected the sprouting and production of the late cultivars CB and FM on the two rootstocks evaluated. The year 2021 experienced typical

rainfall, and the water balance showed a cumulative surplus, enabling the resumption of vegetative growth, subsequent flowering, and fruit set.

The ability of citrus trees to adapt to different environmental fluctuations enables wide geographical distribution, resulting in distinct vegetative and productive performance [30]. The subtropical wetlands of southeast Brazil have highly variable climatic conditions caused by conflicting tropical and subtropical air masses, which result in insufficient water balance in certain years or locations and enhance the risk of citrus flushes with reduced intensities [1].

Under subtropical conditions in south-central São Paulo, citrus flower induction occurs mainly because of low temperatures and water deficits. The citrus flowering period shows a pattern between varietal groups, occurring from June to October for sweet orange trees [1].

Temperature and rainfall, in addition to edaphic conditions, are the main factors that affect growth and determine the entry of citrus trees into the reproductive phase [1]. For FM orange grafted onto nine rootstocks, Carvalho et al. [3] found that the occurrence of new shoots was positively correlated with the increase in minimum and mean temperatures. The results obtained in the present study for this cultivar also confirmed that higher temperatures and a surplus of water favored the beginning of fruit ripening (Figure 1).

The data obtained in this study confirmed the precocity of RL compared to SC, which was reported as the first and greatest flushing of RL. The lower number of days and degree-days of both scions grafted on RL can be attributed to various hypotheses, including increased efficiency in water absorption by RL that accelerates fruit growth [40]. Stenzel et al. [21] evaluated the growth of FM orange fruits on different rootstocks in northern Paraná, Brazil, and concluded that the trees had the earliest reproductive phase when grafted onto RL.

Rootstocks have an important effect on grafted orange trees. It is known that RL induces high vegetative vigor in scions, providing marked differences in canopy height and volume [5,41]. Tree growth in SC is less vigorous than in RL [18]. Studies on the communication between roots and shoots and modifications in scion physiology by rootstocks are very important. Grafting improves tolerance to combined drought and heat stress by modifying the citrus scion's metabolism [12].

The FM had a lower canopy volume than the CB, indicating an aptitude for dense planting, which can increase the yield per area [8]. Azevedo et al. [42] evaluated the production of 'Folha Murcha' grafted on 'Rangpur' lime under different planting spacings and observed that the smallest spacing between plants provided the highest productivity. Smaller trees have some advantages, such as ease of cultivation, phytosanitary control of pests and diseases, and greater efficiency in fruit harvesting [43].

The yield was directly related to the planting stand used. The scion/rootstock combinations evaluated in the 2020 harvest season showed lower performance than expected. This crop failure, ranging from 35 to 50% depending on the region, occurred in all regions of the state of São Paulo. This result can be explained by the negative influence of the climate during the harvests because, during this period, dry spells resulted in high temperatures and water deficits, making it difficult for the newly formed fruits to set [6].

Water scarcity and high temperatures are some of the most adverse climatic conditions worldwide [12]. Drought induces stomatal closure, reduces $CO_2$ intake and photosynthetic activity, increases oxidative stress, and causes membrane damage [19]. Heat stress enhances transpiration and water consumption, alters cell membrane structure, causes protein denaturation and enzyme inactivation, and increases oxidative damage [13].

The results of the productivity index of 3.44 kg cm$^{-2}$ for the combinations evaluated were within the expected standard, as this variable evaluates the CY with the area of the stem of the trees, indicating what was produced in relation to tree growth.

SC is considered to be more susceptible to unfavorable soil and climatic conditions than RL [8]. This may be related to the lower performance observed for this rootstock, as there were intense periods of drought during the study period. The differences between the rootstocks with higher RL yields can be attributed to the quantity and horizontal and



vertical distribution of roots and radicles, water absorption and transport efficiency, and leaf stomata opening and closing [44]. These results suggest that depending on the rootstock, irrigation may be beneficial, as with SC, because a water deficit can have a major influence on fruit setting, output, and quality [30].

In a comparative study between several cultivars suitable for the juice industry, Arenas-Arenas et al. [45] observed that 'Shamoti' sweet orange showed the highest production per kg tree$^{-1}$ among the studied cultivars. According to Nascimento et al. [10], the CB orange trees were more productive than the FM trees, and high-quality fruits could remain on the trees for a long period of time without loss of quality.

The occurrence of alternate bearing in citrus trees is higher in mandarin trees than in orange trees [33]. However, the authors pointed out that studying this alternation is important in sweet orange orchards. By evaluating the performance of FM in a three-year-old orchard and on 12 rootstocks in the northern region of the state of São Paulo, they observed that RL produced the lowest ABI index among the rootstocks. This difference may have occurred because of the edaphoclimatic differences in each region and interactions between the scion and rootstock [14].

The alternate bearing may occur because the trees arrive at flowering with smaller carbohydrates, as they were spent in the previous harvest, thus causing a significant drop in the number of fruits per tree [6]. This factor, combined with the water deficit during the study period, may explain the sharp drop in production observed from the 2019/2021 to the 2020/2021 harvests. Metabolic reconfiguration, including changes in carbohydrate and amino acid fluxes, is a key response for citrus adaptation to a combination of drought and heat stress and highlights the importance of rootstocks in orchards grown under adverse environmental conditions [12].

SC presented greater PE than RL, as it produced smaller growth in the scions. This result indicates that this rootstock is suitable for dense planting, as it induces high production per unit area and a smaller canopy volume than trees grafted onto RL. Cantuarias-Avilés et al. [33] observed that the production efficiencies of FM/RL and FM/SC combinations did not differ significantly, with mean values of 5.32 and 6.86 kg m$^{-3}$, respectively.

The ripening of sweet orange fruits is strongly influenced by the climatic conditions in the growing region [14]. The thermal amplitude directly influenced the CCI and RI. The effect of rootstock on fruit quality is due to different biological processes, such as the induction of gene expression, changes in the activity of several enzymes and hormones [46], and the expression and transport of miRNAs and proteins [15]. Rootstocks can accelerate or delay fruit ripening owing to their effects on tree sprouting [14].

For changes in citrus peel color to occur, chlorophyll must be degraded and carotenoids synthesized. This process is influenced by the climate, in which high- and low-temperature ranges are unfavorable [14]. In the present study, a decline in the CCI occurred from October 2019 to January 2020, a period when high and low temperatures were observed (Figure 1). Such climatic conditions elevate the content of total soluble solids; however, this type of environment induces the production of fruits with greener peels [39]. However, depending on the growing region and harvest, these results may differ from what was expected, thus highlighting the importance of localized research.

The lower external coloration of fruits may be a consequence of low chlorophyll degradation, which may have been stimulated mainly by the temperature under the experimental conditions [47]. The low coloration of citrus fruits is an impediment to the Brazilian export of fresh fruits [48].

During the ripening of CB orange fruits (August to October 2019), the temperature and temperature range remained high, and environmental conditions were considered suitable for chlorophyll degradation and carotenoid synthesis in the fruit peel [39]. The air temperature has the greatest influence on the exterior color of oranges during maturation. Usually, the skin color of fruits grown in colder regions changes later than that of those grown in warmer regions, and, in some cases, the exterior color of the fruits may not develop satisfactorily [49].

The firmness of the fruits decreased throughout the evaluation period because this is related to the cohesion force between the pectins. During maturation, the insoluble pectins are transformed into soluble ones by pectinolytic enzymes, resulting in the softening of the fruits [50].

JY varied from 40 to 55% between the first and last evaluations for both scions evaluated. This is above the 35% required by the juice processing industries [23]. The authors evaluated different orange cultivars on the RL and obtained a JY between 35 and 45% under tropical conditions in the municipality of Rio Branco, Acre, Brazil. The JY and the NS in the fruit of orange trees CB, FM, and 'Valencia' were compared, and the results showed that CB had a lower NS and a higher JY than FM or 'Valencia' [10]. Tazima et al. [51] evaluated the combination of Satsuma 'Okitsu' mandarin trees grafted on nine rootstocks and concluded that trees grafted onto SC produced fruits with higher JY than trees grafted on RL.

The ripening of citrus fruits passes through different phenological stages, and the final stage is characterized by an increase in the TSS content and a decrease in the TA. Harvest quality and optimal citrus harvest time are based on the relationship between SS/TA, that is, the RI, or ratio. The RI represents the balance between sugar and organic acid content in fruits and is associated with juice taste [21].

Citrus is in high demand worldwide as a quality requirement, both for fruits intended for the juice industry and those consumed fresh. The main qualities of the fruits for fresh consumption are the color of the peel, which must be uniform, and a reduced NS, while the yield of juice must be greater than 35% for fruits destined for the juice industry [8]. The fruit must have a content of 10 °Brix or higher to meet the requirements established by the citrus table fruit classification standard [52]. Thus, the use of SC rootstocks is recommended.

In general, the quality of orange juice is determined by the fruit processor. In extracted juice, the sugar concentration typically varies from 9 °Brix for early-season varieties to 12 °Brix for late-season varieties. However, citrus processors usually consider 11.8 °Brix and 11.0 °Brix as the minimum grade of orange juice for frozen and concentrated orange juice (FCOJ) and "Not from concentrate" (NFC), respectively [14]. In the present study, the CB/SC showed an average TSS of 11 °Brix, which is required by the orange juice processing industries in the state of São Paulo, Brazil [53].

Rodrigues et al. [4] studied the performance of the 'Pera' orange tree on different rootstocks and soil and climate conditions, reporting mean TSS values between 8.40 and 9.40 °Brix. Sampaio et al. [53] presented an average of 10.92 °Brix for the fruits of 'Pera' grafted on eight rootstocks under dryland conditions in the region of Cruz das Almas, Bahia, Brazil. Cantuarias-Avilés et al. [33] reported that FM/RL produced fruits with low TSS content, indicating that the juices produced by this combination would have to be blended with juices from other cultivars with higher TSS content.

A good-quality orange fruit should present contents of total soluble solids around 10 °Brix, TA varying from 0.5% to 1%, and RI or a ratio higher than 8 for table fruits and 14 for fruits destined for the production of juice [8]. Despite the unfavorable climatic conditions that occurred during the years in which this study was carried out, the fruit from the CB orange tree reached adequate levels of TSS and RI for fresh consumption and for use in the juice industry. In contrast, the fruit from the FM orange tree presented adequate levels for fresh consumption only. This shows that, although there were losses in the productive variables, the quality of the fruit from the CB orange trees was not affected by the water deficit.

The reduction in TSS content observed in the days after the point of maximum value may be related to the dilution of the juice present in the fruits due to the increase in rainfall that occurred during the period [54]. Similar results were reported by Stenzel et al. [21] in their evaluation of the ripening of FM fruits on different rootstocks; they verified a quadratic decrease in the citric acid content of the fruits. This can be explained by the conversion of organic acids into sugars or their use in respiratory processes [23].

Differences in TA between RL and SC rootstocks were also reported by Gonzatto et al. [55]. Coppice cultivars grafted onto *P. trifoliata* and its hybrids produce fruits with

better qualitative attributes; however, the causes of this phenomenon remain poorly understood [15]. Researchers attribute this beneficial effect to genetically linked response variables, such as the differentiated expression of small RNAs.

These results indicate that the CB cultivar can be an off-season alternative to the traditionally used orange trees in the state of São Paulo, Brazil, since its fruit can be harvested between the harvests of the 'Pera' orange tree and the traditionally used late-ripening orange trees.

AA is an important natural antioxidant and one of the main indicators of orange juice quality [56]. During the ripening of orange fruits, the AA content of the pulp decreases, ranging from 35 to 70 mg 100 mL$^{-1}$ at the time of harvest [14]. The AA present in citrus fruits is directly related to the variety used and the different soil and climatic conditions of the cultivation region [2]. In the present study, the variables TA and AA presented average values of 0.81% and 49.18 mg 100 mL$^{-1}$, respectively, for all evaluated germplasms. These values are within the desirable averages for the production of oranges suitable for fresh consumption and juice processing in São Paulo, Brazil [8].

The data show that the decrease in AA as a function of the evaluation stage may have been influenced by climate. There were periods of severe drought during the assessment period (Figure 1). A possible explanation for the decrease in AA is that variations in AA concentration may occur during fruit development in response to different stressors [8]. Photosynthesis, temperature, and light exposure affect AA synthesis and production [57]. Irradiation and stress stimulate the expression of genes involved in AA [58]. Beber et al. [23] also verified quadratic and linear decreases for the contents of AA in the fruits of orange trees, with average contents varying between 128.02 and 104.73 mg 100 g$^{-1}$.

An increase in the percentage of TSS and a decrease in TA in fruits caused an increase in the RI, or ratio, which was used to determine fruit maturation [2]. As TI includes both fruit weight and SS, the outcome of this variable is influenced by factors that affect the performance of these parameters [59]. A decrease in JY translates into a decline in TI [38].

Stenzel et al. [21] reported that the fruit of FM/RL had an average value of 2.6 kg TSS box$^{-1}$. Cantuarias-Avilés et al. [33] evaluated the performance of FM orange trees grafted on different rootstocks and observed that the RL and SC produced fruits with TI averages of 2.48 and 2.61 kg TSS box$^{-1}$, respectively.

In the present study, the CB/SC presented a TI that met the requirements of the orange juice processing industries. The reduction in TI in the fruits from FM can be explained by the fact that it is calculated using the contents of JY and SS, which are influenced by all the factors that affect these variables.

The presence of many seeds is undesirable for fresh consumption or for processing into orange juice. Fruits with more seeds have a lower market value. CB produced fewer seeds per fruit than FM.

In the marketing of fruits and vegetables, there is a significant difference in price, as measured by the size and quality of the fruits sold in the wholesale market. Common and navel oranges were divided into categories according to equatorial diameter, with Class A fruits having an equatorial diameter greater than 70 mm, Class B fruits having an equatorial diameter between 65 and 70 mm, and Class C fruits having an equatorial diameter smaller than 65 mm [53]. The greater FL and diameter produced by the trees grafted on RL may be attributed to the robustness of this rootstock, which has a deep root system that allows for greater absorption of water and nutrients [41]. Heavier fruits have increasingly diluted organic acids [2] and a lower JY, with fewer carbohydrates, acids, amino acids, and vitamins in the juice [23]. These reports may explain the results obtained, as the SC rootstock favored the development of smaller fruits with higher qualitative attributes.

Fruit ripening of the scions grafted in the RL occurred earlier than that of the SC. The fruits of CB can be harvested in October, whereas the fruits of FM can be harvested in January and February in the hot temperate mesothermal climate of the municipality of São Manuel, in the central-west region of the state of São Paulo, Brazil.

The CB cultivar stood out as an option for the diversification of orchards because of its high yield performance, even under water deficits and high temperatures, and its low index of alternate bearing. In addition, the CB/SC combination produced fruits with the highest CCI, SS, and TI values at the end of ripening.

The trees of both scion cultivars grafted onto RL had a shorter crop cycle, the earliest fruit ripening, and a larger canopy volume. SC produced less alternate bearing in the scions and greater production efficiency, owing to the smaller size of the trees grafted onto it. These results indicate the possibility of increasing the density of the planting stand, promoting productive gain per unit area, and facilitating phytosanitary and cultural practices.

The results obtained may provide additional information for the diversification of germplasm through the selection of new scion/rootstock combinations for citrus orchards, considering the growing region, climatic conditions, thermal requirements, alternate bearing, growth and production performance, fruit maturation, and quality, as well as the dual market demands for processing into juice and for fresh consumption.

**Author Contributions:** Conceptualization, S.L., J.D.F., J.M.A.S. and G.M.N.; methodology, G.M.N., C.P.C., M.A.T., M.L. and S.L.; validation, R.C.M., G.M.N., J.M.A.S., M.L. and C.P.C.; formal analysis, G.M.N.; investigation, G.M.N.; resources, S.L.; data curation, G.M.N.; writing—original draft preparation, G.M.N.; writing—review and editing, S.L. and G.M.N.; visualization, G.M.N.; supervision, S.L.; project administration, S.L.; funding acquisition, S.L., J.D.F. and M.L. All authors have read and agreed to the published version of the manuscript.

**Funding:** This work was partially supported by the Brazilian National Council for Scientific and Technological Development (CNPq), grant numbers 302611/2021-5 and 302848/2021-5.

**Institutional Review Board Statement:** Not applicable.

**Informed Consent Statement:** Not applicable.

**Data Availability Statement:** Data are contained within the article.

**Acknowledgments:** The authors thank the Secretary of Agriculture and Supply of the State of São Paulo for the donation of seedlings through the UAT/CATI agreement.

**Conflicts of Interest:** The authors declare no conflict of interest. The funders had no role in the design of the study; in the collection, analysis, or interpretation of data; in the writing of the manuscript; or in the decision to publish the results.

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
