# Peer review of "Germplasm Diversification in Citrus Orchards in a Mesothermal Climate in Brazil"

_agriculture, doi:10.3390/agriculture13081551_

Round 1

Reviewer 1 Report

Dear author,

Many thanks for your good study and report. My comments are mentioned in the attached file.

Regards

The overall quality of English is good but there are some minor mistakes.

Reviewer 2 Report

Dear Authors, congratulations on your good work. The present study is very interesting to me, at least. It is rich of new information to some extent and has significant practical relevance. In general, it is well-written and discussed.

I listed some comments in the text. They must be taken into account in order to improve the paper's readability. 

My main comments concern the data presentation as a mean of two years. In my opinion, this is a big limit that needs a strong revision. Mainly considering the object of the paper. Please consider some of the suggestions that I have reported in the paper.

War regards

Round 2

Reviewer 2 Report

Dear Editor, I am pleased to send you my report about the second round revision of the paper agriculture-2489292. The authors reported a strict revision according to my suggestion, but these were mainly formal o minor revisions. Concerning the main criticism detected in the paper (the absence of statistical analysis between the two analyzed years) no adjustments were made. I am really sorry, but their response "The addition of harvest-season as a source of variation would make it difficult to discuss the results of the 28 variables evaluated. This is because the study has already highlighted the influence of climatic factors on the results obtained" is unacceptable. I proposed two simple ways to overpass this criticism and no one was taken into account. I want to suggest a third way to obtain discrimination over the years. Authors must include a double-enter table in which the 28 studied variables are reported in columns and the genotypes should be reported above. For each combination, the year-to-year variation should be reported by: ns,*,**, *** accordingly to the ANOVA results. On the other hand, these results are already in the authors' possession!!

Warm regards
